# Asymptotic Homogenization Applied to Flexoelectric Rods

**DOI:** 10.3390/ma12020232

**Published:** 2019-01-11

**Authors:** David Guinovart-Sanjuán, Jose Merodio, Juan Carlos López-Realpozo, Kuppalapalle Vajravelu, Reinaldo Rodríguez-Ramos, Raúl Guinovart-Díaz, Julián Bravo-Castillero, Federico J. Sabina

**Affiliations:** 1Department of Mathematics, University of Central Florida, 4393 Andromeda Loop N, Orlando, FL 32816, USA; Guinovart@ucf.edu (D.G.-S.); kuppalapalle.vajravelu@ucf.edu (K.V.); 2Departamento de Mecánica de los Medios Continuos y T. Estructuras, E.T.S. de Caminos, Canales y Puertos, Universidad Politécnica de Madrid, C.P. 28040 Madrid, Spain; 3Departamento de Matemática, Universidad de La Habana, San Lazaro y L, La Habana 10400, Cuba; jclrealpozo@matcom.uh.cu (J.C.L.-R.); rerora2006@gmail.com (R.R.-R.); guino@matcom.uh.cu (R.G.-D.); 4Instituto de Investigaciones en Matemáticas Aplicadas y en Sistemas, Universidad Nacional Autónoma de México, Alcaldía Álvaro Obregón, Apartado Postal 20-126, CDMX Mexico 01000, Mexico; julian@mym.iimas.unam.mx (J.B.-C.); fjs@mym.iimas.unam.mx (F.J.S.)

**Keywords:** composite, flexoelectric, homogenization method

## Abstract

In this manuscript, the equilibrium problem for a flexoelectric one-dimensional composite material is studied. The two-scales asymptotic homogenization method is used to derive the homogenized formulation of this problem. The manuscript offers a step-by-step methodology to derive effective coefficients and to solve local problems. As an illustrative example, results reported in the literature for piezoelectric composites are obtained as a particular case of the formulation derived here. Finally, three flexoelectric/piezoelectric composites are studied to illustrate the influence of the flexoelectric property on the effective coefficients and the global behavior of the structure.

## 1. Introduction

In the last 35 years, the study of piezoelectric composites has increased because of their use in several areas of engineering [1], contributing to the development of new mechanical structures that are used in high-technology devices [2,3]. The wide range of applications of piezoelectric composites has advanced the development of mathematical, experimental, and computational models related to the study of the properties of these materials, see for instance References [4,5,6]. Piezoelectric composites have been extensively studied, but only in very recent years, flexoelectric materials have taken an important place in the new scientific works, see Reference [7].

Piezoelectricity is usually expressed as an interaction between mechanical strain and one of the electrical variables: the electric field, the electric displacement, or the electric polarization. In Reference [8], the author examines the consequences of considering an additional, linear, electromechanical effect: an interaction between the strain and the polarization gradient. Experiments have shown that when large amplitude mechanical disturbances propagate through a dielectric medium a voltage is developed across the ends of the sample, see Reference [9]. Some of these experiments with nonpiezoelectric elements have been shown to produce a polarization charge upon shock loading. Strain-gradient-induced polarization is known as the flexoelectric effect. According to Reference [10], flexoelectricity is an electromechanical effect of dielectric materials whereby they exhibit a spontaneous electrical polarization induced by a strain gradient (inhomogeneous deformation). The flexoelectric effect can be considered as a high-order electromechanical phenomenon with respect to the piezoelectric effect [7].

Flexoelectricity was first theoretically predicted in the 1950’s and described from a phenomenological standpoint in the 1960’s [11]. Recently, flexoelectric materials have gained prominence due to the development of new methods for manufactured materials with coupled mechanical and electrical behavior [11] and their applications in membrane structures [12,13,14].

In References [15,16], important contributions to the study of flexoelectric materials have been presented highlighting the differences between piezoelectric and flexoelectric structures. The authors emphasize the significance of the flexoelectric effect discussing several applications which include materials with large flexoelectric coefficients as well as cases in which electromechanical coupling of piezoelectricity is not present and in applications related to soft materials. Moreover, some important applications of the flexoelectric effect are focused on the study of biological membranes and the development of piezoelectric structures without using piezoelectric materials, sensing, actuating, or energy harvesting.

Many authors have studied the flexoelectric mechanical equilibrium equations (for instance References [11,17]), bringing together different approaches to solve the problem at hand. One of the most common methods in solid mechanics to approximate the solution of the equilibrium problem of a solid (elastic, piezoelectric, flexoelectric, etc.) is the two-scales asymptotic homogenization method (AHM) [18,19]. The multiscale asymptotic homogenization method has been widely used to derive the effective properties of composite materials with different mechanical properties. In References [18,20,21,22], the AHM is used to derive the effective properties of elastic composites (laminates, fibrous, wavy, etc.). These ideas are extended to the study of piezoelectric structures [23]; viscoelastic composites [24]; and thermo-piezoelectric materials [25]. To the best of the authors knowledge, a methodology to find the effective properties of flexoelectric materials has not been presented before. For that reason, this work acts to extend those methodologies (that find effective properties) to include the case of flexoelectric materials using the two-scales asymptotic homogenization method, which provides the first step to subsequently study three-dimensional curvilinear flexoelectric structures.

In general, the two-scale asymptotic homogenization method is used to find the effective properties of periodic composites, i.e., for any mechanical property p(x), $x\in R^{n}$, there is a $T\in R^{n}$ s.t. p(x+T)=p(x) for all $x\in R^{n}$. For non-periodic media, statistical and numerical methods have been developed to find representative volume elements and the associated effective properties [26,27,28]. In Reference [29], the self-consistent method in explicit form (effective field) and implicit form (effective medium) as well as the AHM are used to derive the effective piezoelectric properties of fibrous composites with randomly positioned fibers and periodically distributed fibers, showing good agreement among the different methods.

The work is organized as follows. In Section 2, a theoretical framework for the equilibrium of a one-dimensional periodic flexoelectric composite material is presented. The asymptotic expansions of the displacement and the electric potential are introduced in Section 3, using the two-scales asymptotic homogenization method, and a methodology to derive the expression of the effective coefficients depending on the so-called local function expressions is illustrated. The process to obtain the analytic solutions of the local problems is shown. A variational formulation of the local problem is proposed and the system is solved using the classical finite element method (FEM). As an important benchmark, the model reproduces the solution of the local problems reported in Reference [30] for a one-dimensional piezoelectric structure. Finally, in Section 4, the theoretical results derived in the previous sections are used to study some numerical cases. The effective coefficients reported in References [19,30] for a two-material piezoelectric structure made of lead zirconate titanate (PZT) C91 and P-82 are derived using the model developed here, which makes use of the solution of the local problem variational formulation (FEM). In order to extend the study for the case of randomly distribute one-dimensional flexoelectric composites, a two-constituents structure $\Omega_{N}$ is considered for which the materials properties are randomly distributed along the rod with a binomial probability. Using the concept of correlation length for the particular case of one-dimensional binomial distributed structure, as reported in Reference [31], the methodology is then extended to the case of non-periodic structures. As it is shown in Reference [29], the effective properties of random composites converge to the effective properties of periodic structures when the length of $\Omega_{N}$ increases. In addition, the combination of flexoelectric/piezoelectric/non-piezoelectric composites are studied in order to illustrate the influence of the flexoelectricity on the effective properties. One of the composites is considered as the combination of two flexoelectric materials, barium titanate (BaTiO${}_{3}$) and strontium titanate (SrTiO${}_{3}$) [32,33,34]. Other composites are combinations of these two flexoelectric materials with a exclusively piezoelectric material, polyvinylidene fluoride (PVDF) [35,36], a non-piezoelectric polymer described in Reference [37] and PZT-7A reported in Reference [38]. From these material combinations important conclusions are derived with respect to the influence of the flexoelectric property in the global behavior of the composites. Finally, a comparison between the solutions of a heterogeneous and homogeneous problem for a one-dimensional flexoelectric structure is presented. The finite element method is used to solve the problems of two different examples considering the same prescribed boundary conditions but different external forces.

## 2. Flexoelectric One-Dimensional Problem

A one-dimensional flexoelectric composite rod Ω is considered. The constitutive relations between stress σ and electrical displacement *D* with strain ϵ and electric field *E* are written in the following form (see Reference [17]),
(1)$$\sigma \left(x\right)=C\left(x\right)\epsilon \left(x\right)-e\left(x\right)E\left(x\right)+\mu \left(x\right)\frac{dE(x)}{dx},$$
(2)$$D\left(x\right)=e\left(x\right)\epsilon \left(x\right)+\mu \left(x\right)\frac{d\epsilon (x)}{dx}+\kappa \left(x\right)E\left(x\right),$$
where C(x), e(x), κ(x), and μ(x) denote the stiffness, piezoelectric, permittivity, and flexoelectric tensors, respectively. Linear deformations of the composite, the strain, and the electric field are studied in terms of the displacement *u* and the electric potential ϕ as
(3)$$\epsilon \left(x\right)=\frac{du(x)}{dx},$$
(4)$$E\left(x\right)=-\frac{d\phi (x)}{dx}.$$

Assume the material parameters *C*, *e*, κ, μ are rapidly oscillating periodic functions along the rod with respect to the variable *y*, i.e.,
$$C\left(y\right)\equiv C\left(\frac{x}{\varepsilon }\right),\;\;e\left(y\right)\equiv e\left(\frac{x}{\varepsilon }\right),$$
$$\kappa \left(y\right)\equiv \kappa \left(\frac{x}{\varepsilon }\right),\;\;\mu \left(y\right)\equiv \mu \left(\frac{x}{\varepsilon }\right),$$
where y=x/ε∈Y is the local variable, since 0<ε<<1 is a very small parameter that characterizes the periodicity of the structure, and *Y* is the periodic cell. The composite has the structure presented in Figure 1.

To simplify the equations, material parameters and functions are considered to depend on the position variable *x* or local variable *y*, i.e., u≡u(x), ϕ≡ϕ(x), C≡C(y), e≡e(y), κ≡κ(y), μ≡μ(y). The equilibrium equations presented in Reference [17] for one-dimensional solids take the following expression:(5)$$\frac{d\sigma }{dx}+f=\frac{d}{dx}\left(C\frac{du}{dx}+e\frac{d\phi }{dx}-\mu \frac{d^{2}\phi }{dx^{2}}\right)+f=0\;\;\mathrm{in}\;\;\Omega ,$$
(6)$$\frac{dD}{dx}=\frac{d}{dx}\left(e\frac{du}{dx}+\mu \frac{d^{2}u}{dx^{2}}-\kappa \frac{d\phi }{dx}\right)=\rho \;\;\mathrm{in}\;\;\Omega ,$$
with boundary conditions
(7)$$\begin{array}{c} u\left(0\right)=u_{0},\;\;\frac{du}{dx}\left(0\right)=w_{0},\;\;\phi \left(0\right)=\varphi_{0}, \end{array}$$
(8)$$\begin{array}{c} \sigma \left(1\right)=S_{1},\;\;\mu \left(1\right)\frac{dE}{dx}\left(1\right)=-r_{1},\;\;D\left(1\right)=-\tau_{1}, \end{array}$$
where the functions *f* and ρ are the external forces and body charge density, respectively. On the other hand, the values $u_{0}$, $S_{1}$, $w_{0}$, $r_{1}$, $\varphi_{0}$, and $\tau_{1}$ are the prescribed displacement, traction, displacement derivative, high-order traction, electric potential and charge, respectively. Perfect contact conditions at the interface are considered, i.e., function displacement *u* is differentiable and classical stress Cϵ+eE, higher order stress $\mu \frac{dE}{dx}$ (see Reference [17]), electric potential ϕ and electric displacement *D* are considered continuous functions at the interfaces of the materials. The boundary value problem (Equation 5)–(Equation 8) is cast as a linear system of third order ordinary differential equations for which, under the perfect contact conditions as mentioned above, existence and uniqueness of the solution on the interval [0,1] is guaranteed [39,40].

## 3. Asymptotic Homogenization Method

The boundary value problem (Equation 5)–(Equation 8) has rapidly oscillating coefficients. In order to approach the solution of the problem, an equivalent homogenized problem must be obtained. To derive the expression of the homogenized system, the two-scales asymptotic homogenization method is used. In Reference [19], the asymptotic expansion to the solution of the piezoelectric problem is reported as
(9)$$u\left(x,y\right)=v_{0}+\sum_{k=1}^{\infty }\varepsilon^{k}\left[N_{k}\left(y\right)\frac{d^{k}v_{0}}{dx^{k}}+\Pi_{k}\left(y\right)\frac{d^{k}\phi_{0}}{dx^{k}}\right],$$
(10)$$\phi \left(x,y\right)=\phi_{0}+\sum_{k=1}^{\infty }\varepsilon^{k}\left[\Psi_{k}\left(y\right)\frac{d^{k}v_{0}}{dx^{k}}+\Theta_{k}\left(y\right)\frac{d^{k}\phi_{0}}{dx^{k}}\right],$$
where $v_{0}\left(x\right)$ and $\phi_{0}\left(x\right)$ only depend on the global variable *x* and the functions $N_{k}$, $\Pi_{k}$, $\Psi_{k}$, and $\Theta_{k}$ are ε-periodic continuous functions that only depend on the local variable *y*, i.e., $N_{k}\left(y+\varepsilon \right)=N_{k}\left(y\right)$, $\Pi_{k}\left(y+\varepsilon \right)=\Pi_{k}\left(y\right)$, $\Psi_{k}\left(y+\varepsilon \right)=\Psi_{k}\left(y\right)$, and $\Theta_{k}\left(y+\varepsilon \right)=\Theta_{k}\left(y\right)$.

Now, the expansions (Equation 9)–(Equation 10) are introduced into the equilibrium Equations (Equation 5)–(Equation 6), for which one needs to consider the derivative operator
$$\frac{du}{dx}=u_{x}+\frac{1}{\varepsilon }u_{y}\;\;\;,\;\frac{d\phi }{dx}=\phi_{x}+\frac{1}{\varepsilon }\phi_{y}\;\;\;,$$
where ${\left(\cdot \right)}_{x}$ and ${\left(\cdot \right)}_{y}$ denote the partial derivative with respect to *x* and *y*, respectively. After some simple manipulations, (Equation 5) and (Equation 6) take the following expressions:(11)$$\begin{array}{c} \sum_{k=-1}^{\infty }\sum_{m=1}^{\infty }\varepsilon^{k}\left[L_{km}\frac{d^{m}v_{0}}{dx^{m}}+P_{km}\frac{d^{m}\phi_{0}}{dx^{m}}\right]+f=0, \end{array}$$
(12)$$\begin{array}{c} \sum_{k=-1}^{\infty }\sum_{m=1}^{\infty }\varepsilon^{k}\left[Q_{km}\frac{d^{m}v_{0}}{dx^{m}}+R_{km}\frac{d^{m}\phi_{0}}{dx^{m}}\right]=\rho , \end{array}$$
respectively, where $L_{km}$, $P_{km}$, $Q_{km}$, and $R_{km}$ denote the coefficient of the *m* derivative of $v_{0}$ and $\phi_{0}$ that multiplies $\varepsilon^{k}$. For instance, $L_{-11}$ is the coefficient of the first derivative of $v_{0}$ that multiplies $\varepsilon^{-1}$. As a result of the continuity requirements, when ε→0, the coefficients of k=−1 are equated to zero. The non-identical zero coefficients are the so-called local problems. In what follows, we characterize these problems.

### 3.1. Local Problems

The following local problems have to be considered. The system of local problems (LQ):(13)$$\begin{array}{c} L_{-11}\equiv \frac{d}{dy}\left(C+C\frac{dN_{1}}{dy}+e\frac{d\Psi_{1}}{dy}\right)=0, \end{array}$$
(14)$$\begin{array}{c} L_{-12}\equiv -\frac{d}{dy}\left(\mu \frac{d^{2}\Psi_{2}}{dy^{2}}+2\mu \frac{d\Psi_{1}}{dy}\right)-\mu \frac{d^{2}\Psi_{1}}{dy^{2}}=0, \end{array}$$
(15)$$\begin{array}{c} Q_{-11}\equiv \frac{d}{dy}\left(e+e\frac{dN_{1}}{dy}-\kappa \frac{d\Psi_{1}}{dy}\right)=0, \end{array}$$
(16)$$\begin{array}{c} Q_{-12}\equiv \frac{d}{dy}\left(\mu +\mu \frac{d^{2}N_{2}}{dy^{2}}+2\mu \frac{dN_{1}}{dy}\right)+\mu \frac{d^{2}N_{1}}{dy^{2}}=0, \end{array}$$
with interface conditions
$$\left[\;\left[C+C\frac{dN_{1}}{dy}+e\frac{d\Psi_{1}}{dy}\right]\;\right]=\left[\;\left[\mu \frac{d^{2}\Psi_{2}}{dy^{2}}+2\mu \frac{d\Psi_{1}}{dy}\right]\;\right]=\left[\;\left[e+e\frac{dN_{1}}{dy}-\kappa \frac{d\Psi_{1}}{dy}\right]\;\right]=\left[\;\left[\mu +\mu \frac{d^{2}N_{2}}{dy^{2}}+2\mu \frac{dN_{1}}{dy}\right]\;\right]=0,$$
where $\left[\;\left[\cdot \right]\;\right]={\left(\cdot \right)}_{\left(+\right)}-{\left(\cdot \right)}_{\left(-\right)}$ denotes the jump at the interface.

Furthermore, the system of local problems (PR):(17)$$\begin{array}{c} P_{-11}\equiv \frac{d}{dy}\left(e+C\frac{d\Pi_{1}}{dy}+e\frac{d\Theta_{1}}{dy}\right)=0, \end{array}$$
(18)$$\begin{array}{c} P_{-12}\equiv -\frac{d}{dy}\left(\mu +\mu \frac{d^{2}\Theta_{2}}{dy^{2}}+2\mu \frac{d\Theta_{1}}{dy}\right)-\mu \frac{d^{2}\Theta_{1}}{dy^{2}}=0, \end{array}$$
(19)$$\begin{array}{c} R_{-11}\equiv -\frac{d}{dy}\left(\kappa +\kappa \frac{d\Theta_{1}}{dy}-e\frac{d\Pi_{1}}{dy}\right)=0, \end{array}$$
(20)$$\begin{array}{c} R_{-12}\equiv \frac{d}{dy}\left(\mu \frac{d^{2}\Pi_{2}}{dy^{2}}+2\mu \frac{d\Pi_{1}}{dy}\right)+\mu \frac{d^{2}\Pi_{1}}{dy^{2}}=0, \end{array}$$
with interface conditions
$$\left[\;\left[e+C\frac{d\Pi_{1}}{dy}+e\frac{d\Theta_{1}}{dy}\right]\;\right]=\left[\;\left[\mu +\mu \frac{d^{2}\Theta_{2}}{dy^{2}}+2\mu \frac{d\Theta_{1}}{dy}\right]\;\right]=\left[\;\left[\kappa +\kappa \frac{d\Theta_{1}}{dy}-e\frac{d\Pi_{1}}{dy}\right]\;\right]=\left[\;\left[\mu \frac{d^{2}\Pi_{2}}{dy^{2}}+2\mu \frac{d\Pi_{1}}{dy}\right]\;\right]=0.$$

The LQ system is derived from the coefficients multiplying the derivatives of $v_{0}$ in each equation and it relates the local functions $N_{1}$, $\Psi_{1}$, $N_{2}$, and $\Psi_{2}$. Similarly, the PR system is obtained from the coefficients of the derivatives of $\phi_{0}$ and it relates the local functions $\Theta_{1}$, $\Pi_{1}$, $\Theta_{2}$, and $\Pi_{2}$.

### 3.2. Methodology to Solve Local Problems Using a System of Linear Equations

To obey conditions (13)–(20), the LQ and PR systems need to be solved. More specifically, the functions $\frac{dN_{1}}{dy}$, $\frac{d\Psi_{1}}{dy}$, $\frac{d^{2}N_{2}}{dy^{2}}$, $\frac{d^{2}\Psi_{2}}{dy^{2}}$, $\frac{d\Pi_{1}}{dy}$, $\frac{d\Theta_{1}}{dy}$, $\frac{d^{2}\Pi_{2}}{dy^{2}}$, and $\frac{d^{2}\Theta_{2}}{dy^{2}}$ need to be obtained. Later, we show that the local functions $N_{1}$, $N_{2}$, $\Psi_{1}$, $\Psi_{2}$, $\Pi_{1}$, $\Pi_{2}$, $\Theta_{1}$, and $\Theta_{1}$ are not necessary to find the homogenized problem.

The system of local problems is solved using ideas described in Reference [20]. Here, we summarize the methodology. Let us focus on the LQ system. Integrating Equations (13) and (15) the following system is derived:$$\left(\begin{array}{cc} C & e\\ e & -\kappa \end{array}\right)\left(\begin{array}{c} \frac{dN_{1}}{dy}\\ \frac{d\Psi_{1}}{dy} \end{array}\right)=\left(\begin{array}{c} \lambda_{1}-C\\ \lambda_{3}-e \end{array}\right),$$
where $\lambda_{\alpha }\equiv \lambda_{\alpha }\left(x\right)$, α=1,3. Thus, the solutions of the system are
(21)$$\frac{dN_{1}}{dy}=\frac{\kappa }{e^{2}+C\kappa }\lambda_{1}+\frac{e}{e^{2}+C\kappa }\lambda_{3}-1,$$
(22)$$\frac{d\Psi_{1}}{dy}=\frac{e}{e^{2}+C\kappa }\lambda_{1}-\frac{C}{e^{2}+C\kappa }\lambda_{3}.$$

To find expressions for $\lambda_{\alpha }$ in terms of the materials parameters *C*, *e*, κ and μ, one needs to use the average operator with respect to the local variable. This average operator is given by
(23)$$\langle \cdot \rangle =\frac{1}{V_{Y}}\int_{Y}\left(\cdot \right)dy,$$
where $V_{Y}=\left|Y\right|$. From the periodicity of $N_{1}$ and $\Psi_{1}$ and taking the average operator in both sides of the Equations (21) and (22), the following system for $\lambda_{\alpha }$ is obtained:(24)$$\langle \frac{\kappa }{e^{2}+C\kappa }\rangle \lambda_{1}+\langle \frac{e}{e^{2}+C\kappa }\rangle \lambda_{3}=1,$$
(25)$$\langle \frac{e}{e^{2}+C\kappa }\rangle \lambda_{1}-\langle \frac{C}{e^{2}+C\kappa }\rangle \lambda_{3}=0.$$

It follows that $\lambda_{1}$ and $\lambda_{3}$ are solutions of the system (24) and (25). Considering *C*, *e*, κ, and μ to be constant functions along each material, one gets that $\frac{d^{2}N_{1}}{dy^{2}}=\frac{d^{2}\Psi_{1}}{dy^{2}}=0$. Then, Equations (14) and (16) are transformed into
(26)$$\frac{d}{dy}\left(\mu \frac{d^{2}\Psi_{2}}{dy^{2}}+2\mu \frac{d\Psi_{1}}{dy}\right)=0,$$
(27)$$\frac{d}{dy}\left(\mu +\mu \frac{d^{2}N_{2}}{dy^{2}}+2\mu \frac{dN_{1}}{dy}\right)=0,$$
respectively. Finally, solutions of the LQ system are given by (21), (22) and
(28)$$\frac{d^{2}N_{2}}{dy^{2}}=-\frac{2k}{\Delta }\lambda_{1}-\frac{2e}{\Delta }\lambda_{3}+\frac{1}{\mu }\lambda_{4}+1,$$
(29)$$\frac{d^{2}\Psi_{2}}{dy^{2}}=-\frac{2e}{\Delta }\lambda_{1}+\frac{2C}{\Delta }\lambda_{3}+\frac{1}{\mu }\lambda_{2},$$
where $\Delta =e^{2}+C\kappa$ and $\lambda_{i}$, i=1,2,3,4, satisfy the system of equations
(30)$$\left(\begin{array}{cccc} \langle \kappa /\Delta \rangle & 0 & \langle e/\Delta \rangle & 0\\ -2\langle \kappa /\Delta \rangle & 0 & -2\langle e/\Delta \rangle & \langle 1/\mu \rangle \\ \langle e/\Delta \rangle & 0 & -\langle C/\Delta \rangle & 0\\ -2\langle e/\Delta \rangle & \langle 1/\mu \rangle & 2\langle C/\Delta \rangle & 0 \end{array}\right)\left(\begin{array}{c} \lambda_{1}\\ \lambda_{2}\\ \lambda_{3}\\ \lambda_{4} \end{array}\right)=\left(\begin{array}{c} 1\\ -1\\ 0\\ 0 \end{array}\right).$$

Similarly, solutions of the PR system (17)–(20) are obtained. Expressions for these solutions are given in Appendix A.

### 3.3. Variational Formulation of the LQ System. Finite Element Method (FEM)

The finite element method is a strong and convenient numerical tool to solve systems of ordinary or partial differential equations. It is an alternative method to approximate the solution for the LQ systems (13)–(16). This is the procedure. Let us take Y=[0,γ)∪[γ,1], where γ∈(0,1). Consider the test functions $v_{i}$, i = 1, 2, 3, 4, multiplying the LQ problem. Integrating along *Y*, using integration by parts and taking into account the continuity of the function at the interface, the following variational formulation for the LQ system is derived:(31)$$\begin{array}{c} \int_{Y}\left(C+C\frac{dN_{1}}{dy}+e\frac{d\Psi_{1}}{dy}\right)\frac{\partial v_{1}}{\partial y}dy=0, \end{array}$$
(32)$$\begin{array}{c} \int_{Y}\left(\mu \frac{d^{2}\Psi_{2}}{dy^{2}}+2\mu \frac{d\Psi_{1}}{dy}\right)\frac{\partial v_{2}}{\partial y}dy=0, \end{array}$$
(33)$$\begin{array}{c} \int_{Y}\left(e+e\frac{dN_{1}}{dy}-\kappa \frac{d\Psi_{1}}{dy}\right)\frac{\partial v_{3}}{\partial y}dy=0, \end{array}$$
(34)$$\begin{array}{c} \int_{Y}\left(\mu +\mu \frac{d^{2}N_{2}}{dy^{2}}+2\mu \frac{dN_{1}}{dy}\right)\frac{\partial v_{4}}{\partial y}dy=0. \end{array}$$

Periodic conditions for the local functions and perfect contact at the interface are considered. Now, using Equations (31)–(34) and the standard finite element method [41] on the domain *Y*, the local functions are approximated.

### 3.4. Effective Coefficients and Homogenized Problem

To find the effective coefficients, the average operator is used on the coefficients of $\varepsilon^{0}$ in Equations (11) and (12), i.e., one deals with $L_{0m}$, $P_{0m}$, $Q_{0m}$, and $R_{0m}$. Taking into account the periodicity of the local functions as well as *C*, μ, *e*, and κ, the non-identical zeros are
(35)$$\begin{array}{c} \langle L_{02}\rangle =\langle C+C\frac{dN_{1}}{dy}+e\frac{d\Psi_{1}}{dy}\rangle , \end{array}$$
(36)$$\begin{array}{c} \langle L_{03}\rangle =\langle 2\mu \frac{d\Psi_{1}}{dy}+\mu \frac{d^{2}\Psi_{2}}{dy^{2}}\rangle , \end{array}$$
(37)$$\begin{array}{c} \langle P_{02}\rangle =\langle e+C\frac{d\Pi_{1}}{dy}+e\frac{d\Theta_{1}}{dy}\rangle , \end{array}$$
(38)$$\begin{array}{c} \langle P_{03}\rangle =\langle \mu +2\mu \frac{d\Theta_{1}}{dy}+\mu \frac{d^{2}\Theta_{2}}{dy^{2}}\rangle . \end{array}$$
(39)$$\begin{array}{c} \langle Q_{02}\rangle =\langle e+e\frac{dN_{1}}{dy}-\kappa \frac{d\Psi_{1}}{dy}\rangle , \end{array}$$
(40)$$\begin{array}{c} \langle Q_{03}\rangle =\langle \mu +2\mu \frac{dN_{1}}{dy}+\mu \frac{d^{2}N_{2}}{dy^{2}}\rangle , \end{array}$$
(41)$$\begin{array}{c} \langle R_{02}\rangle =\langle \kappa +\kappa \frac{d\Theta_{1}}{dy}-e\frac{d\Pi_{1}}{dy}\rangle , \end{array}$$
(42)$$\begin{array}{c} \langle R_{03}\rangle =\langle 2\mu \frac{d\Pi_{1}}{dy}+\mu \frac{d^{2}\Pi_{2}}{dy^{2}}\rangle . \end{array}$$

Substituting the expressions of the solutions of the LQ and PR systems, which are given in (21), (22), (28), (29), and the Appendix Equations (Equation 63)–(Equation 66), into the Equations (35)–(42), the following relations are obtained
$$\langle L_{03}\rangle \equiv \langle R_{03}\rangle \equiv 0;\;\langle P_{03}\rangle \equiv \langle Q_{03}\rangle ;\;\langle P_{02}\rangle \equiv \langle Q_{02}\rangle .$$

Therefore, the effective coefficients of Equations (11) and (Equation 12) are
(43)$$\begin{array}{c} \hat{C}=\langle C+C\frac{dN_{1}}{dy}+e\frac{d\Psi_{1}}{dy}\rangle , \end{array}$$
(44)$$\begin{array}{c} \hat{e}=\langle e+C\frac{d\Pi_{1}}{dy}+e\frac{d\Theta_{1}}{dy}\rangle \equiv \langle e+e\frac{dN_{1}}{dy}-\kappa \frac{d\Psi_{1}}{dy}\rangle , \end{array}$$
(45)$$\begin{array}{c} \hat{\kappa }=\langle \kappa +\kappa \frac{d\Theta_{1}}{dy}-e\frac{d\Pi_{1}}{dy}\rangle , \end{array}$$
(46)$$\begin{array}{c} \hat{\mu }=\langle \mu +2\mu \frac{d\Theta_{1}}{dy}+\mu \frac{d^{2}\Theta_{2}}{dy^{2}}\rangle \equiv \langle \mu +2\mu \frac{dN_{1}}{dy}+\mu \frac{d^{2}N_{2}}{dy^{2}}\rangle . \end{array}$$

The original problem (Equation 5) and (Equation 6), by means of the homogenization technique, is transformed into the following homogenized problem (in terms of the effective coefficients):(47)$$\frac{d}{dx}\left(\hat{C}\frac{dv_{0}}{dx}+\hat{e}\frac{d\phi_{0}}{dx}-\hat{\mu }\frac{d^{2}\phi_{0}}{dx^{2}}\right)+f=0\;\;\mathrm{in}\;\;\Omega ,$$
(48)$$\frac{d}{dx}\left(\hat{e}\frac{dv_{0}}{dx}+\hat{\mu }\frac{d^{2}v_{0}}{dx^{2}}-\hat{\kappa }\frac{d\phi_{0}}{dx}\right)=\rho \;\;\mathrm{in}\;\;\Omega ,$$
(49)$$\begin{array}{c} v_{0}\left(0\right)=u_{0},\;\;\frac{dv_{0}}{dx}\left(0\right)=w_{0},\;\;\phi_{0}\left(0\right)=\varphi_{0}, \end{array}$$
(50)$$\begin{array}{c} \sigma_{0}\left(1\right)=S_{1},\;\;\hat{\mu }\left(1\right)\frac{E_{0}}{dx}\left(1\right)=r_{1},\;\;D_{0}\left(1\right)=-\tau_{1}, \end{array}$$
where $\sigma_{0}$, $E_{0}$, and $D_{0}$ are the effective stress, effective electric field, and effective electrical displacement, respectively.

## 4. Analysis of Numerical Results

### 4.1. Model Validation with the Particular Case of Piezoelectric Materials

To validate the model, our goal is to reproduce the results of the one-dimensional piezoelectric composite analyzed in Reference [30]. The constituents of this one-dimensional piezoelectric structure are lead zirconate titanate (PZT) C-91 and P-82. The materials properties are given in Table 1 of Reference [30]. For this particular case, the flexoelectric coefficient μ associated with the components of the structure is considered zero. Therefore, the equilibrium problems (Equation 5)–(Equation 8) take the expression (1) and (2) of Reference [30]. Using the asymptotic expansion (Equation 9) and (Equation 10) and following the methodology described in the previous section, the non-vanishing local problems are (13), (15), (18), and (20), which corresponds with Equations (23) and (24) in Reference [30]. Finally, the expressions of the effective coefficients for this piezoelectric composite are given by Equations (43), (44), and (45).

In Table 1, the values reported for the effective coefficients $\hat{C}$, $\hat{e}$, $\hat{\kappa }$ of this piezoelectric composite considered in References [19,30] are compared with the results obtained using the present model (Section 3) and the results obtained using FEM (solving the variational formulation of the LQ and PR systems). The volume fraction of C-91, for the numerical calculation, is considered to be $V_{1}\in$(0, 0.4, 0.8, 1). To avoid singularities of solutions of LQ and PR systems, the value of flexoelectric property for the piezoelectric composites is taken as $\mu =10^{-20}$. Suffice to say that the results reported in Reference [19,30] are reproduced using both the AHM (present model) and FEM (variational formulation) methods. Therefore, the piezoelectric composite results are obtained as a particular case of the flexoelectric composite formulation developed here.

### 4.2. Flexoelectric Composite Rod without Ideal Periodicity

It is interesting to study the the above-described methodology for the case of random composites, i.e., composites without periodicity. A two-elements flexoelectric rod where the constituents are $L_{1}$ (BaTiO${}_{3}$) and $L_{2}$ (SrTiO${}_{3}$) with a fixed length of 1 unit is considered. Assume $\Omega_{N}=\left[0,N\right]$, for every 0≤n<N, n∈N, the region [n,n+1] is occupied by the layer $L_{1}$ with probability p=0.7146. The numbers $N_{1}$ and $N_{2}$ represent the amount of components $L_{1}$ and $L_{2}$ in $\Omega_{N}$, respectively, and $N=N_{1}+N_{2}$. The average operator is considered
(51)$${\langle \cdot \rangle }_{N}=\frac{1}{N}\int_{\Omega_{N}}\left(\cdot \right)dx=\frac{N_{1}}{N}{\left(\cdot \right)}^{\left(1\right)}+\frac{N_{2}}{N}{\left(\cdot \right)}^{\left(2\right)}.$$
This average operator is a particular case of the Equation (2.2.1) of Reference [31] for one-dimensional structures. From the results described in Section 3 and considering the average operator (51), the effective properties (43)–(46) are easily computed for a structure without periodicity. To compare the behavior of the effective properties for random composites with the ones for periodic structures, a periodic composite Ω is considered. The constituents of Ω are BaTiO${}_{3}$ with volume fraction p=0.7146 and SrTiO${}_{3}$ with volume fraction 1−p. In Table 2, the material properties of BaTiO${}_{3}$ and SrTiO${}_{3}$ are given. Figure 2 illustrates the behavior of three experiments with random composites considering a probability p=0.7146 for $L_{1}$ and a composite with periodicity and a volume fraction of p=0.7146 for BaTiO${}_{3}$ for $N\in [1,2^{25}]$. The three random composites were randomly generated with MATLAB using the so-called rand function.

Figure 2 shows that the values of effective properties for non-periodic composites coincide with the values of the effective coefficients for a periodic structure when the number of layers increase. It follows that the methodology is valid for structures with no periodicity, making it possible to approximate the effective properties for random composites considering the periodic structures. Similar results are obtained in Reference [42], for the particular case of an elastic Fibonacci laminate composite.

### 4.3. Influence of the Flexoelectric Property on the Effective Coefficients

In this section, the influence of the flexoelectric property on the effective coefficients is studied. A bi-material one-dimensional composite is considered. In Table 2, the properties of the flexoelectric materials barium titanate (BaTiO${}_{3}$) and strontium titanate (SrTiO${}_{3}$), the active piezoelectric materials polyvinylidene fluoride (PVDF) and PZT-7A, and soft non-piezoelectric polymer are shown. To illustrate the effective coefficients of flexoelectric/piezoelectric structures, several combinations of flexoelectric composite with other well-known piezoelectric materials are considered.

In Figure 3, the influence of the flexoelectricity in combination with an active polymer PVDF which has piezoelectric properties on the effective characteristics is studied using three composites. The first composite is made of the flexoelectric material BaTiO${}_{3}$ and the piezoelectric active polymer PVDF; the second one is the composition of the flexoelectric composite SrTiO${}_{3}$ and PVDF; the materials of the third composite are the piezoelectric constituents PZT-7A and PVDF (see Reference [38]). With the values given in Table 2 and the methodology described in Section 3, the effective coefficients for each composite are obtained. Figure 3 illustrates the behavior of the effective coefficients for different volume fractions of the composites. The effective coefficient $\hat{C}$ continues increasing for the three combinations of materials. In addition, as is shown in the figure, the influence of the flexoelectric materials on the piezoelectric and the dielectric effective coefficients is remarkable. For both BaTiO${}_{3}$-PVDF and SrTiO${}_{3}$-PVDF, the piezoelectric effective coefficient $\hat{e}$ increases until $V_{1}=0.9$ and $V_{1}=0.8$, respectively, from which the property decays. On the other hand, for PZT-PVDF, the piezoelectric effective tensor decreases for all values of $V_{1}$. Similarly to $\hat{C}$, the dielectric effective coefficient $\hat{\kappa }$ increases with the volume fraction for the three combinations of materials, although for the composites formed by a flexoelectric material this property is greater than that of the PZT-PVDF composite. The effective flexoelectric coefficient $\hat{\mu }$ varies only for the composites with a flexoelectric component (either BaTiO${}_{3}$ or SrTiO${}_{3}$).

To study the influence of non-flexoelectric and non-piezoelectric constituents in the effective properties for composites with a flexoelectric component, three structures are studied. One of the structures is the combination of the flexoelectric materials BaTiO${}_{3}$ and SrTiO${}_{3}$; the constituents for the second structure are SrTiO${}_{3}$ and a non-piezoelectric polymer; the last structure is made of strontium titanate (SrTiO${}_{3}$) and a piezoelectric material PZT-7A. In Figure 4, the effective properties of these three composites for different volume fractions are shown. An important detail to highlight is that for all the effective properties in Figure 4, the values of the effective coefficients of the SrTiO${}_{3}$-polymer are lower than the values of the effective coefficients of the other two combinations. The non-piezoelectric polymer properties diminish the values of the composite effective coefficients, compared with the other composites. On the other hand, the combination of the two flexoelectric materials reinforce the values of $\hat{C}$, $\hat{e}$, $\hat{\kappa }$, and $\hat{\mu }$, compared with the composite made of the non-piezoelectric polymer. It is shown in Figure 4 that the effective piezoelectric property $\hat{e}$ is almost zero for the combination of SrTiO${}_{3}$ with a non-piezoelectric polymer. A similar situation is shown for the flexoelectric parameter, i.e., $\hat{e}$ remains close to zero for the composites that involve a non-flexoelectric component (either SrTiO${}_{3}$-polymer or SrTiO${}_{3}$-PZT). Therefore, it can be concluded that for composites made of flexoelectric material with non-flexoelectric material, the global behavior of the effective coefficients of the structure is similar to a non-flexoelectric material. A parallel situation occurs when considering the composition of a piezoelectric with a non-piezoelectric structure as it is illustrated in Figure 4. Hence, for composites made of a piezoelectric material with a non-piezoelectric material the global behavior of the effective coefficients of the structure is similar to a non-piezoelectric material.

The dependence of the effective coefficients $\hat{C}$, $\hat{e}$, and $\hat{\kappa }$ on the local functions $\frac{dN_{1}}{dy}$, $\frac{d\Psi_{1}}{dy}$, $\frac{d\Theta_{1}}{dy}$, $\frac{d\Pi_{1}}{dy}$ is shown in Equations (43)–(45). From Equations (21), (22), the Appendix Equations (Equation 63), (Equation 65), and the solutions of the systems (30), (Equation 67), the non-influence of the μ parameter in the local functions is derived. Therefore, the effective coefficients $\hat{C}$, $\hat{e}$, and $\hat{\kappa }$ are not affected by the flexoelectric component in the case of non-flexoelectric materials. From the homogenized problem (47)–(50), it can be followed that the influence of the effective flexoelectric component in the effective stress and electric displacement cannot be underestimated.

The investigation of the flexoelectric effect goes beyond the theoretical analysis of the materials properties to include real measurements [43]. Recent papers have established detailed descriptions of their experiments to determine the flexoelectric properties of PVDF. More precisely, References [44] and [45] describe experiments to find the effective flexoelectric components $\mu_{1123}$ and $\mu_{2312}$ of the fourth order tensor $\mu_{ijkl}$ for polyvinylidene fluoride. On account of the great difficulty in obtaining some experimental measurements, the relationship between the strain gradient and torque is deduced theoretically and further verified with finite element analysis. The approach is applied to test responses in bars machined from bulk polyvinylidene fluoride [45]. On the other hand, in Reference [46], the flexoelectricity of prototypical semicrystalline polymer, α-phase PVDF, films are investigated. The paper presents a step-by-step description of the experiment highlighting the direct flexoelectric effect in the α-phase PVDF films.

### 4.4. Solution of the Heterogeneous and Homogenized Problems

In this section, the homogenized problem (47)–(50) is solved for a flexoelectric one-dimensional structure. In order to compare the approximation between the solution of the heterogeneous problem and the homogenized problem, the finite element method is used. Consider Ω a two-element structure made of barium titanate (BaTiO${}_{3}$) and strontium titanate (SrTiO${}_{3}$) periodically distributed along Ω. In order to illustrate the example, the prescribed conditions for the heterogeneous (Equation 5)–(Equation 8) and the homogeneous (47)–(50) problems are given as follows: $u_{0}=1$, $w_{0}=1$, $\varphi_{0}=0$, $S_{1}=0$, $r_{1}=0$, and $\tau_{1}=0$. Two different volume fractions for BaTiO${}_{3}$ are considered $V_{1}=0.5$, case 1, and $V_{1}=0.8$, case 2. The external forces are taken as $f\left(x\right)=e^{-x}$ for case 1 and f(x)=sin(x) for case 2. The body charge density ρ is considered identically zero along Ω. The values of the effective coefficients of the homogenized problem are computed using the methodology described in Section 3 and are shown in Table 3.

In Figure 5, a comparison of the stress obtained from the solution of the heterogeneous problem (Equation 5)–(Equation 8) and the homogeneous equivalent problem (47)–(50) for ε=1/5 is shown. As a result of the continuity of the coefficients of the homogeneous problem, the stress function has a smooth behavior along Ω. On the other hand, the discontinuity of the coefficients in the heterogeneous problem leads to the obstacles in the solution, see Reference [47].

## 5. Conclusions

In this manuscript, the equilibrium equation for a flexoelectric composite has been studied. The two-scales asymptotic homogenization method is used to find the homogenized problem formulation. The work presents the expressions of the effective coefficients for the case of a flexoelectric structure. The local problems are derived along with their corresponding variational formulations. The manuscript offers a detailed description to solve the local problems for one-dimensional structures. The effective coefficients reported in References [19,30] for piezoelectric composite are computed here as particular cases of both the present model (Section 3) and the finite element method. The methodology is used to find the effective properties of random structures. The results illustrate the convergence of the effective coefficients for random structures to the case of a composite with ideal periodicity. In addition, the influence of the flexoelectric parameter on the global behavior of composites has been studied. The effective properties of three bi-materials (composites) made of combinations of BaTiO${}_{3}$, SrTiO${}_{3}$, PVDF, a soft non-piezoelectric polymer, and PZT-7A were obtained. The influence of the flexoelectric parameter (μ) on the effective stress and electric displacement was shown. As an example, a comparison between the solutions of the heterogeneous and homogeneous problem for a flexoelectric structure is presented. A good approximation between the solutions can be appreciated when ε is zero. The results presented in this manuscript can be extended to more complex structures with wavy behavior and three-dimensional composites.

## Figures and Tables

**Figure 1 materials-12-00232-f001:**
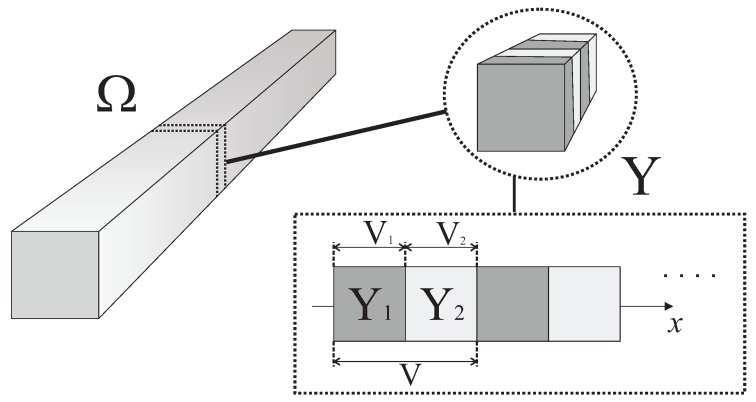
One-dimensional flexoelectric rod and the corresponding periodic cell.

**Figure 2 materials-12-00232-f002:**
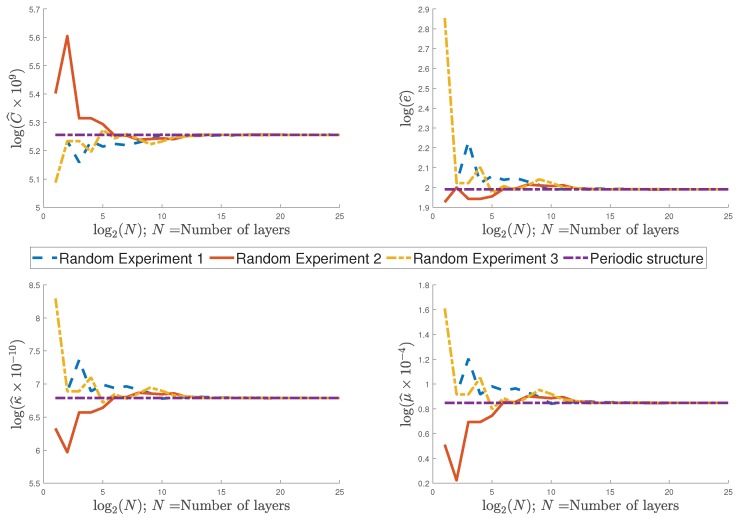
Effective properties and structures of three random flexoelectric composites with ideal periodicity.

**Figure 3 materials-12-00232-f003:**
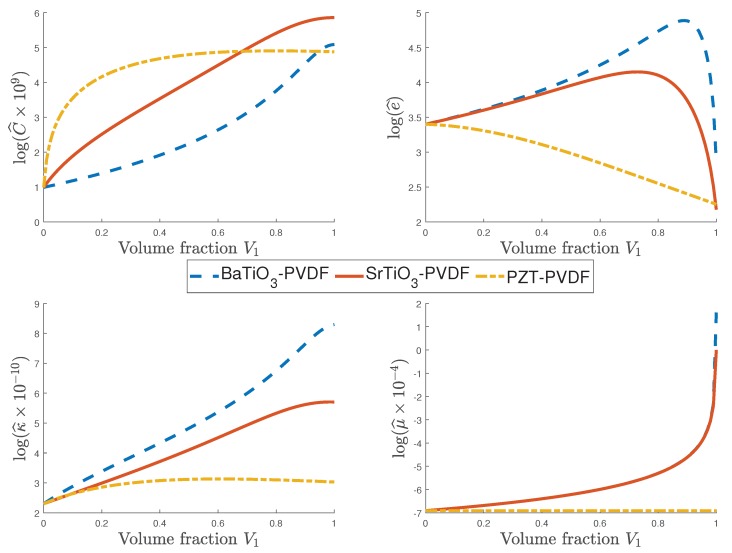
Values of different effective coefficients for three composites: i) flexoelectric material BaTiO${}_{3}$ and the piezoelectric active polyvinylidene fluoride (PVDF) polymer; ii) composition of the flexoelectric composite SrTiO${}_{3}$ and PVDF; and iii) the piezoelectric constituents PZT-7A and PVDF vs. values of volume fraction.

**Figure 4 materials-12-00232-f004:**
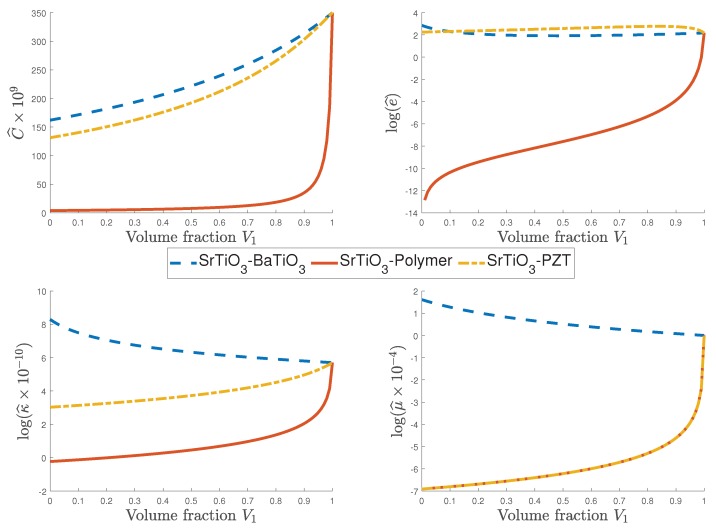
Values of different effective coefficient of three composites made of the combination of two flexoelectric material SrTiO${}_{3}$-BaTiO${}_{3}$, a non-piezoelectric polymer with SrTiO${}_{3}$, and piezoelectric PZT-7A with a flexoelectric material SrTiO${}_{3}$ vs. values of volume fraction.

**Figure 5 materials-12-00232-f005:**
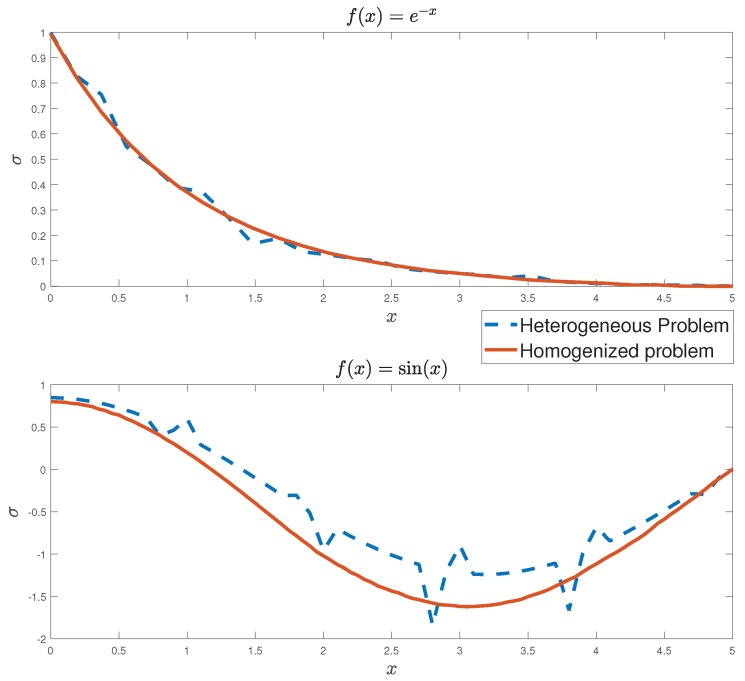
Values of the stress function σ, solutions of the heterogeneous (Equation 5)–(Equation 8) and homogeneous (47)–(50) problems for different external forces functions: $f\left(x\right)=e^{-x}$ and f(x)=sin(x).

**Table 1 materials-12-00232-t001:** Comparison of the effective coefficients $\hat{C}$, $\hat{e}$, $\hat{\kappa }$ for a piezoelectric composite computed using the methodology described in Section 3, the finite element method (FEM), and the results reported in References [19,30].

Effective Coefficient $\hat{C}$ (10${}^{9}$ N/m${}^{2}$)				
$V_{1}$	**Present Model**	**FEM**	**Ref. [30]**	**Ref. [19]**
0.0	118.300	118.300	118.299	118.300
0.4	116.953	116.953	116.952	116.953
0.8	115.401	115.401	115.400	115.401
1.0	114.500	114.500	114.499	114.500
**Effective Coefficient** $\hat{e}$ **(C/m** ${}^{2}$ **)**				
0.0	26.400	26.400	26.399	26.400
0.4	24.806	24.806	24.805	24.806
0.8	22.634	22.863	22.633	22.634
1.0	21.200	21.200	21.199	21.200
**Effective Coefficient** $\hat{\kappa }$ **(10** ${}^{-10}$ **F/m** **)**				
0.0	110.900	110.900	110.899	110.900
0.4	148.056	148.056	148.054	148.056
0.8	200.835	200.836	200.832	200.835
1.0	236.600	236.600	236.595	236.600

**Table 2 materials-12-00232-t002:** Material properties of barium titanate (BaTiO${}_{3}$), strontium titanate (SrTiO${}_{3}$), PVDF, non-piezoelectric polymer, and PZT-7A are given. *C*, *e*, κ, and μ denote the elastic, piezoelectric, dielectric and flexoelectric tensors, respectively. The materials with μ=0 are called non-flexoelectric (PVDF, polymer, PZT-7A).

Materials	*C* (10${}^{9}$ N/m${}^{2}$)	*e* (C/m${}^{2}$)	κ (10${}^{-10}$ F/m)	μ (10${}^{-4}$ C/m)
BaTiO${}_{3}$	162 [32]	17.36 [33]	4000 [32]	5 [32]
SrTiO${}_{3}$	350 [32]	8.82 [34]	300 [32]	1 [32]
PVDF	2.7 [35]	30 [36]	10 [36]	0
Polymer	3.86 [37]	0 [37]	0.7965 [37]	0
PZT-7A	131.4 [38]	9.522 [38]	0.372 [38]	0

**Table 3 materials-12-00232-t003:** Coefficients of the homogenized flexoelectric composite made of barium titanate (BaTiO${}_{3}$) and strontium titanate (SrTiO${}_{3}$) for $V_{1}=0.5$ and $V_{1}=0.8$.

Materials	$\hat{C}$ (10${}^{9}$ N/m${}^{2}$)	$\hat{e}$ (C/m${}^{2}$)	$\hat{\kappa }$ (10${}^{-10}$ F/m)	$\hat{\mu }$ (10${}^{-4}$C/m)
$V_{1}=0.5$	221.8664	6.8629	560.04	1.6667
$V_{1}=0.8$	181.8374	8.0288	1158.11	2.7778

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
