# Peer review of "Asymptotic Homogenization Applied to Flexoelectric Rods"

_materials, 2019, doi:10.3390/ma12020232_

Reviewer 1 Report

In the paper a two-scale asymptotic homogenization method for flexoelectric one-dimensional periodic material is proposed and the overall constitutive tensors are determined.

The topic is very interesting and is completely within the scope of Materials.

Suggestions to the Authors:

 1)      The Authors have not adequately analyzed the state of art about the topic treaty in this paper. Several paper dealing the homogenization of periodic flexoelectric materials and multi-field asymptotic homogenization in periodic thermo-piezoelectric materials are published in prestigious journals but none of these works are analyzed and considered here;

2)      An effective benchmark test of the homogenization model should be performed. The solution of the homogenized model should be compared with that obtained from the heterogeneous model;

3)      The Authors should pay more attention to the graphical resolution of the figures;

4)      The Authors must proof check their manuscript for some typos in the text and in the formulas. 

Author Response

We thank the Reviewer 1 for the time spent in reading the manuscript and for the helpful comments. We corrected all suggested typing errors.

1.     Many authors have studied the flexoelectric mechanical equilibrium equations (for instance [11] and [17]), bringing different approaches to solve the problem at hand. One of the most common methods in solid mechanics to approximate the solution of the equilibrium problem of a solid (elastic, piezoelectric, flexoelectric, etc.) is the two-scales Asymptotic Homogenization Method (AHM), ([18], [19]). The multiscale Asymptotic Homogenization Method have been widely used to derive the effective properties of composite materials with different mechanical properties. In papers like [18], [20], [21], [22] the AHM is used to derive the effective properties of elastic composites (laminates, fibrous, wavy, etc.). These ideas are extended to the study of piezoelectric structures, (see [23]); viscoelastic composites (see [24]) thermo-piezoelectric materials (see [25]). To the best of the authors knowledge, a methodology to find the effective properties of flexoelectric materials have not been presented before. For that reason, this work pretends to extend the methodologies to find the effective properties to the case of flexoelectric materials using the two-scales Asymptotic Homogenization method, bringing the first step to study the three-dimensional curvilinear flexoelectric structures.

2.     As an important benchmark, the model reproduces the solution of the local problems reported in [30] for a one-dimensional piezoelectric structure as a particular case. Finally, in Section 4, the theoretical results derived in the previous sections are used to study some numerical cases. The effective coefficients reported in [30] and [19] for a two-material piezoelectric structure made of lead zirconate titanate (PZT) C91 and P-82 are derived using the model developed here, which makes use of the solution of the local problem variational formulation (FEM). In this sense, Table 1 was included as a comparison with the results reported in [30]. Besides, subsection 4.4 “Solution of the heterogeneous and homogenized problems” was included to compare the solution of the heterogeneous and homogenized problem

3.     The resolution of the figures was improved

Some typos in the text and in the formulas have been corrected

Reviewer 2 Report

The authors proposed a homogenization method for deriving effective properties of composite flexoelectric rods. Their approach and the results seem to be technically sound for me. They give illlustrative examples based on real materials. The paper is mathematically oriented. In general, its OK so. However, potential readers are probably also material scientists and physicisits. I could suggest to the authors to try to get their paper more easily accesible to those groups of potential readers, for example by introducing a Discussion Section and explaining how they results can be verified in experiments and what are the design rules for making effective flexoelectric structures.

I have some questions with respect to the content.

You assumed that the material parameters C, e, k, m are rapidly oscillating periodic functions along the rod with respect to the variable y. Will your approach also work for random composites? If it is not possible to reproduce the ideal periodicity, what does it mean for effective properties? How sensitive is your approach with respect to the loss of ideal periodocity.

I am familiar with the concept of the correlation length (see, e.g. Snarskii, A. A., Bezsudnov, I. V., Sevryukov, V. A., Morozovskiy, A., & Malinsky, J. (2007). Transport processes in macroscopically disordered media. From Mean Field Theory to Percolation, Springer, New York). How the correlation length is related to your approach? Is it the spatial period? Please address this issues in your paper.

Author Response

We thank the Reviewer 2 for the positive comments and suggestions for the improvement of the manuscript. Find below the answers to the review comments.

1.     The investigation of the flexoelectric effect goes beyond the theoretical study of the materials properties, [43]. Recent papers have established detailed descriptions of their experiments to determine the flexoelectric properties of PVDF. More precisely, the works [44] and [45] describe experiments to find the effective flexoelectric components μ1123 and μ2312 of the fourth order tensor μijkl for polyvinylidene fluoride. Due to the difficulty of some experimental measurements, the relationship between the strain gradient and torque is deduced theoretically and further verified with finite element analysis and the approach is applied to testing bars machined from bulk polyvinylidene fluoride, [45]. On the other hand, in [46], the flexoelectricity of a prototypical semicrystalline polymer,α−phase PVDF, films are investigated. The letter presents a step-by-step description of the experiment. Highlighting, the giant direct flexoelectric effect in the α-phase PVDF film.

2.     To answer this, a subsection 4.2 “Flexoelectric composite rod without ideal periodicity” was added.

(…) n order to extend the results for the case of randomly distribute one-dimensional flexoelectric composites, a two-constituents structure ΩN is considered, whose the materials properties are randomly distributed along the rod with a binomial probability.  Using the concept of correlation length for the particular case of one-dimensional binomial distributed structure, reported in [31], the methodology is extended to the case of non-periodic structures. As it happened in [29], the effective properties of random composites converge to the effective properties of periodic structures when the length of ΩN increases

Reviewer 3 Report

There is no definition of flexoelectric. Please explain what is it.

Please provide some information why it is important and when it should be considered.

Author Response

We thank the anonymous Reviewer 3 for the positive comments and the suggestions for the improvement. Below are our replies.

1.     Piezoelectricity is usually expressed as an interaction between mechanical strain and one of the electrical variables: the electric field, the electric displacement or the electric polarization. In [8], the author examines the consequences of considering an additional, linear, electromechanical effect: an interaction between the strain and the polarization gradient. Experiments have shown that when large amplitude mechanical disturbances propagate through a dielectric medium a voltage is developed across the ends of the sample, [9]. Some of this experiments with nonpiezoelectric elements have shown to produce a polarization charge upon shock loading. This effect is known as flexoelectric effect.  According to [10], the flexoelectricity is an electromechanical effect of dielectric materials whereby they exhibit a spontaneous electrical polarization induced by a strain gradient (inhomogeneous deformation). In other words, the flexoelectric effect can be considered as a high-order electromechanical phenomenon with respect to the piezoelectric effect, [7].

2.     In [15] and [16], important contributions to the study of flexoelectric materials have been presented highlighting the differences between piezoelectric and flexoelectric structures. The authors emphasize into the significance of the flexoelectric effect and how important it may turn out to be. For example: for materials where the flexoelectric coefficient is really large, when the electromechanical coupling of piezoelectricity is not present or to study soft materials. Some applications of the flexoelectric effect to highlight are the study of biological membranes, to create piezoelectric structures without using piezoelectric materials, sensing, actuating or energy harvesting.   

Round  2

Reviewer 1 Report

The paper is greatly improved by the authors. I think that the article is interesting and deserves publication in Materials.